# Assessment of Vaccination Status in Professional Football Players in Low Categories in Greece

**DOI:** 10.3390/jfmk7040073

**Published:** 2022-09-26

**Authors:** Dimitrios Papagiannis, Georgios Marinos, Ioannis Anyfantis, Georgios Rachiotis

**Affiliations:** 1Public Health & Vaccines Laboratory, Faculty of Nursing, School of Health Sciences, University of Thessaly, 41110 Larissa, Greece; 2Department of Hygiene, Epidemiology and Medical Statistics, School of Medicine, National and Kapodistrian University of Athens, 11527 Athens, Greece; 3European Agency for Safety and Health at Work (EU-OSHA), Santiago de Compostela 12, 48003 Bilbao, Spain; 4Department of Hygiene and Epidemiology, Faculty of Medicine, School of Health Sciences, University of Thessaly, 42200 Larissa, Greece

**Keywords:** football players, vaccines, chief medical officers

## Abstract

Background: There are limited data on the vaccination practices of footballers in low professional categories in Greece and Europe. The aim of this study was to investigate the vaccination practices followed by the medical staff of the low professional football categories in Central Greece. Methods: A questionnaire was developed and included questions on the vaccination practices of football players. The participants were chief medical officers of the fifteen low-category football teams in Central Greece. Overall, 10 out of 15 chief medical officers participated (response rate: 67%). Results: All participants recommended only the tetanus monovalent vaccine in cases with deep trauma of athletes. Influenza, pneumococcal vaccine, and Hepatitis A and B vaccines were not recommended by the medical officers. This was also the case for measles, mumps, rubella (MMR) vaccine and meningococcal vaccine with strains A, C, Y, W. Participants reported the lack of specific guidelines for vaccination in professional footballers. Conclusions: The recent study showed poor vaccination practices in low categories of professional football teams in Central Greece. The qualitative view of the respondents revealed the importance of the absence of guidelines on vaccination of football players.

## 1. Introduction

Professional football players are exposed to physical interaction with other persons, and also, due to the nature of their job, are engaged in travel activities. Consequently, this occupational group may be at risk of infectious diseases [1,2]. On the other hand, there is evidence from surveillance data that the incidence of injuries is higher than the incidence of illness among professional footballers [3]. However, it has been pointed out that even mild diseases could have negative impact on the performance of professional footballers and also prevent them from participating in official football matches [4]. Moreover, even minor infections can reduce the footballer’s ability to sustain heavy training [4,5].

The Union of European Football Associations (UEFA) published official regulations and suggested that every professional football player must have a complete vaccination record according to the national vaccination program of the country where the football player works [6].

There is sparse information on the vaccination practices of high-level professional footballers, and to the best of our knowledge, there is no relevant information on the vaccination practices related to low-category professional footballers.

We conducted a descriptive, cross-sectional study aiming to explore the vaccination practices of the chief medical officers (CMOs) of all low-category (Greek Football League, Second and Third Division) football clubs in Central Greece.

## 2. Materials and Methods

A questionnaire was developed and included questions on the vaccination practices of football players. Two CMOs were used for pilot testing of the questionnaire. Face-to-face interviews were used for pilot testing and data collection. The participating chief medical officers (CMOs) of the fifteen low-category football teams in Central Greece (Greek Football League, Second and Third Division) were asked to report on their vaccination recommendation practices regarding footballers. In particular, they were asked to report whether they recommended vaccination against diphtheria, tetanus, pertussis, polio, pneumococcal, seasonal influenza, hepatitis B, hepatitis A, measles–mumps–rubella, meningitis A, C, Y, W (options: yes/no). In addition, participants were asked to provide an answer to the “qualitative” aspect of the questionnaire, which comprised the following open-ended question: “In your opinion, what are the most important problems related to the assessment of footballers’ vaccination status?”. The 10 football teams included players from around the world; the total number of players participating in this study was 268. The majority of them were Greek, and the number of players coming from countries other than Greece was 42. Data were tabulated, and descriptive statistics were used for presentation of the results. In particular, the absolute (n) and relative frequencies (%) were presented. Statistical analysis was performed using the Excel software. All participants gave their informed consent for participation in the study, and all responses were anonymized.

## 3. Results

Overall, 10 out of the 15 CMOs (response rate: 66.5%) participated in the survey. All participating physicians (100%) recommended a vaccine against Tetanus (Table 1). Influenza, and Hepatitis A and B vaccines were not recommended (Table 1). This was also the case for Diphtheria, pertussis and measles–mumps–rubella vaccines. In addition, vaccines against meningitis A, C, Y, W strains for adults were also not recommended. The qualitative feedback from the participants revealed the lack of guidelines for vaccination in teams and professional footballers. The lack of vaccination guidelines for footballers and gaps related to access to the vaccination history of footballers were the main problems reported by the participating CMOs.

## 4. Discussion

In the present study, we reported a unanimous recommendation for the Tetanus vaccine and no recommendation for the other vaccines recommended by the Greek National Immunization Program for adults. These data contradict the findings of a recent Greek survey conducted among the chief medical officers of Super League Greece. In fact, this study reported a prevalence of recommendation for vaccination varying from 12% for Diphtheria, Tetanus, Pertussis and Polio to 87% for seasonal influenza [7]. In addition, this finding contradicts the results provided by an Italian study from Serie-A, which reported an influenza vaccination rate of 40% [2].

Hepatitis A and B, Diphtheria, Pertussis, Polio and Measles–Mumps–Rubella are recommended vaccines, according to the Greek National Immunization Program, and both vaccines should be given at preschool age. Perhaps this may be the reason why the medical staff of the teams did not recommend these vaccines. Consequently, we cannot exclude the probability that footballers may have been vaccinated against these infectious agents during childhood. Nevertheless, this important information is not included in our database. In particular for Hepatitis B, as an occupational risk to footballers, it should be noted that previous researchers reported a lack of evidence for transmission among Australian footballers [8]. On the contrary, Constantini and colleagues recommended vaccination against Hepatitis B because high-level athletes were considered as a population at risk of various infections, given the nature of their work [9]. Furthermore, it has been suggested that professional athletes should receive all the vaccinations recommended for the general population [10]. Lastly, Signorelli and coworkers suggested specific immunization practices, including a check for childhood and adolescence immunizations, as well as routine immunizations, including influenza (annually) and decennial booster doses for tetanus and diphtheria [11].

Our study has several limitations. The study was questionnaire based, and information bias may have occurred. The regional cross-sectional design is not representative of all teams in Football Leagues I and II in Greece. With respect to the non-respondents (response rate = 67%), we acknowledge the possible occurrence of selection bias, since we were unable to obtain the information from non-respondents.

The inability of the study group to confirm the vaccination history of each athlete individually is another limitation of the study. In addition, we investigated the vaccine recommendation practices, and we were unable to verify the actual vaccination coverage among the players or to report on the seroprevalence levels of vaccinations. Lastly, we acknowledge that not addressing the question of why CMOs would not recommend routine vaccinations is an important limitation of the present study. This topic should be addressed in future research.

## 5. Conclusions

A present study showed poor vaccination practices in low categories of professional football teams in Central Greece. These findings are in marked contrast to a recent survey among the chief medical officers of Super League Greece, which demonstrated better vaccination practices than the present study. Further work is needed in order to update the vaccination practices for footballers of low professional category in Greece. Moreover, the present study revealed the importance of the absence of guidelines on vaccination of football players. Lastly, our findings provide useful insights regarding the immunization practices of football players in low professional categories in Greece and would help policy makers adopt a comprehensive approach to the vaccination of professional footballers.

## Figures and Tables

**Table 1 jfmk-07-00073-t001:** Recommendations for vaccination in Football League II.

Vaccines–Antigens	Teams n%		Teams n%	
	Yes		No	
Diphtheria			10/10	100%
Tetanus	10/10	100%		
Pertussis			10/10	100%
Polio			10/10	100%
Pneumococcal			10/10	100%
Seasonal influenza			10/10	100%
Hepatitis B			10/10	100%
Hepatitis A			10/10	100%
Measles–Mumps–Rubella			10/10	100%
Meningitis A, C, Y, W			10/10	100%

## Data Availability

Not applicable.

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
