# Peer review of "Assessment of Vaccination Status in Professional Football Players in Low Categories in Greece"

_jfmk, 2022, doi:10.3390/jfmk7040073_

Round 1

Reviewer 1 Report

Manuscript titled "Assessment of vaccination status in professional football players in low categories in Greece" by Dimitrios et al is a nice piece of work. I think this work is important and other sports group can also get benefited from this work. The work become even more important in times when global travel, mass gathering and frequent spread if infectious entity is becoming more and more common.

Before acceptance of this work for publication I have few minor suggestion

1) Need minor english editing

2) Better to include sample of questionnaire in the paper

Author Response

We would like to thank the reviewer for the constructive comments to help us to improve the quality of manuscript .

Reviewer 2 Report

Papagiannis et al., in their communication entitled “Assessment of vaccination status in professional football players in low categories in Greece” has focused on the vaccination status among Greece Football players. It is really very focused work and such work are required for the society.

The manuscript is well present but there are few suggestions for the improving of this short communication.

My suggestions/comments are:

1.     Why some of the common words initial is in capital letters in abstract section, like in line no 19 “Chief Medical Officers”, line no 21 “Tetanus”, line no 17 “Football”? Please rectify.

2.     In the Abstract section line 23 “MMR” abbreviation is used directly without explaining it for the first time. Please expand the abbreviation for the very first time.

3.     Some of the sentences are not formatted or justified like line no 36, 62, 75, 85, 86, 88 and 109. Please rectify.

4.     Please add rationale of this study.

Author Response

(The authors gave the same response as above.)

Reviewer 3 Report

Please review my comments. I compliment the authors on their attempt to address an important general public health issue in sports medicine

Author Response

(The authors gave the same response as above.)

Round 2

Reviewer 2 Report

I recommend the manuscript for publication in its current form. The authors have incorporated all the comments suggested mam.

Reviewer 3 Report

The reviewer thanks the authors for their thoughtful consideration of my comments and suggestions and for their incorporation of the revisions. This is an important sport/ health topic and I congratulate the authors for their contribution